# Capacitance Characteristics and Breakdown Mechanism of AlGaN/GaN Metal–Semiconductor–Metal Varactors and Their Anti-Surge Application

**Chien-Fu Shih [1], Yu-Li Hsieh [1,2], Liann-Be Chang [3,4,5,6,*] , Ming-Jer Jeng [1,5], Zi-Xin Ding [1] and Shao-An Huang [6]**

[1] Department of Electronic Engineering, Chang Gung University, Guishan, Taoyuan 333, Taiwan; D9828201@cgu.edu.tw (C.-F.S.); D0527103@cgu.edu.tw (Y.-L.H.); mjjeng@mail.cgu.edu.tw (M.-J.J.); M0827102@cgu.edu.tw (Z.-X.D.)

[2] Department of Electrical and Electronic Engineering, Chung Cheng Institute of Technology, National Defense University, Daxi, Taoyuan 335, Taiwan

[3] Green Technology Research Center, Chang Gung University, Guishan, Taoyuan 333, Taiwan

[4] Department of Materials Engineering, Ming Chi University of Technology, Taishan, New Taipei City 243, Taiwan

[5] Department of Otolaryngology-Head and Neck Surgery, Chang Gung Memorial Hospital, Linkou, Taoyuan 333, Taiwan

[6] Graduate Institute of Electro-Optical Engineering, Chang Gung University, Guishan, Taoyuan 333, Taiwan; M0424011@cgu.edu.tw

* Correspondence: liann@mail.cgu.edu.tw; Tel.: +886-3-211-8800 (ext. 5793); Fax: +886-3-211-8507

**Abstract:** The AlGaN/GaN materials with a wide band gap, high electron mobility, and high breakdown voltage are suitable for manufacturing high-power and high-frequency electronic devices. In this study, metal Schottky contact electrodes of different dimensions are prepared on AlGaN/GaN wafers to fabricate metal–semiconductor–metal (MSM) varactors. Voltage-dependent capacitance and breakdown voltages of the varactors are measured and studied. The corresponding breakdown mechanisms of varactors with different electrode gaps are proposed. Furthermore, an anti-surge application using GaN-based MSM varactors in a signal transmission module is demonstrated, and its surge suppression capability is shown. We believe that our study will be beneficial in developing surge protection circuits for RF applications.

**Keywords:** AlGaN/GaN; metal–semiconductor–metal; varactors; breakdown; anti-surge

## 1. Introduction

The advent of the Internet of Things has placed difficult frequency and power requirements on electronic devices. GaN has been widely used in high-frequency and high-power applications. It has a wide band gap (3.4 eV), high electron mobility, and high breakdown voltage [1]. Because of these distinct advantages, it is more widely used in manufacturing devices for high-power and high-frequency applications compared to other materials [2] (Table 1). Moreover, advances in quantum confinement technology have led to the development of heterostructures of two-dimensional electron gas (2DEG), which can solve the serious problem of decreased channel electron mobility. Consequently, high-electron-mobility field-effect transistors (HEMT) made of AlGaN/GaN are extensively used in high-frequency electronic components [3–5].

It is predicted that remote-controlled, unmanned vehicles will become very popular in the near future. Their safety and reliability are important issues that need to be addressed. The main challenges in unmanned vehicle product reliability are their global positioning system (GPS) positioning and navigation capabilities. Although GPS has the advantage of worldwide, all-weather, and automatic detection, the radar or signal-receiving system is vulnerable to intentional electromagnetic interference/pulse (IEMI/IEMP) attacks, which can cause the navigation system to fail. Furthermore, if the electronic modules are damaged, the whole system is lost permanently. In view of this, the development of corresponding protection components is necessary and has high application potential [6,7].

Anti-surge components that are widely used in protection circuits include gas discharge tubes (GDTs), transient voltage suppressor (TVS) diodes, and metal oxide varistors (MOVs). The GDT has an excellent capability of shunting surge current. However, it takes time (in microseconds) for it to start shunting because the $SF_6$ gas must be ionized first. This time is approximately a few microseconds. Therefore, it is not suitable for responding to attacks with extremely short rise time (i.e., nanoseconds), such as malicious electromagnetic pulses (MEMP). TVS diodes and MOVs have short response time, but for a single component, the shunt ability is relatively limited. This cannot be enhanced because the parasitic capacitance will be too high to be used for high-frequency circuits. One possible candidate is the Darlington current limiter [8]. However, it has a complicated circuit structure and high associated cost.

The GaN-based varactor is manufactured using a GaN wafer. It has a two-dimensional electron gas (2DEG) within it. One of its characteristics is that its capacitance changes along with the applied bias voltage. This component has been widely studied and reported [9–15]. We have proposed GaN-based 2DEG MSM varactors as the anti-surge components connected in series between the antenna and the backend circuits, as shown in Figure 1 [16], and we have demonstrated that this design protects the back-end HEMT element from electrostatic discharge (ESD) [17]. In this study, we further apply this protection concept to MEMP attacks. It is known that the gate of the advanced silicon-based field-effect transistor has a breakdown voltage of 30 V. If the GaN-based varactor is connected in series at its front end, the overall circuit robustness can be improved. However, the requirement is that the varactor must have a rapid response to the attack surge and a high breakdown voltage value. Under normal working conditions, the varactor capacitance must be high enough to transmit the signal by reducing the insertion loss. As the high-energy pulse attack happens, its capacitance must immediately decrease to stop the attack energy from invading the back-end circuit. In terms of the manufacturing process, it must also be compatible with GaN-based high-frequency components [18]. Therefore, the development of related components using AlGaN/GaN epitaxial structure materials with 2DEG to protect the GPS instruments has a very high potential.

In this regard, investigation into the influence of the capacitance conversion ratio and breakdown voltage of varactors is very important. To the best of our knowledge, such investigation has not been reported in published literature. In this study, metal Schottky contact electrodes of different dimensions are designed and prepared on AlGaN/GaN wafers to produce metal–semiconductor–metal (MSM) varactors. Not only the voltage-dependent capacitance, but also the breakdown voltage and mechanism of the designed varactors are investigated. Finally, the signal transmission performance and anti-surge ability are presented.

**Table 1.** Characteristics of semiconductor materials for high-power applications.

| Material Properties | Si | SiC | GaAs | GaN |
|---|---|---|---|---|
| Energy gap (eV) | 1.11 | 3.25 | 1.43 | 3.49 |
| Thermal conductivity (W cm$^{-1}$ K$^{-1}$) | 1.5 | 4.9 | 0.56 | 1.5 |
| Electron mobility (cm$^2$ V$^{-1}$ s$^{-1}$) | 1350 | 700 | 8500 | 2000 |
| Hole mobility (cm$^2$ V$^{-1}$ s$^{-1}$) | 450 | 120 | 330 | 300 |
| Breakdown voltage (MV cm$^{-1}$) | 0.25 | 3.5 | 0.4 | 4 |
| Saturation electron speed (10$^7$ cm s$^{-1}$) | 1 | 2.1 | 1.3 | 2.5 |
| Dielectric constant | 11.9 | 10 | 12.5 | 9 |

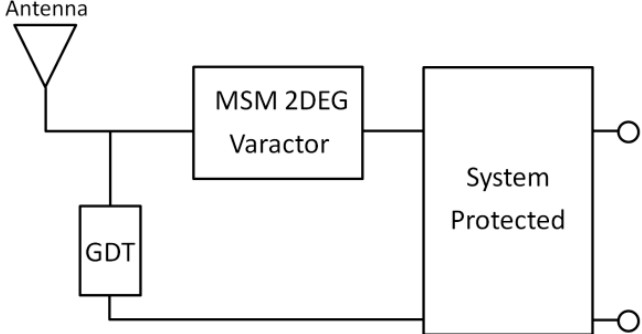

**Figure 1.** Schematic diagram of the structure of the surge protection module, in which the metal–semiconductor–metal (MSM) varactor is placed in series.

## 2. Experiment

### 2.1. Fabrication of the GaN-Based 2DEG MSM Varactor

First, the mask of the MSM varactors was designed. Six different component lengths (2000, 1500, 1000, 500, 250, and 150 µm) were used for the patterns in the mask. Next, for each specific electrode length, six different gap widths (30, 25, 20, 15, 10, and 5 µm) were also designed, as shown in Figure 2a. The wafer used to fabricate devices in this experiment was an AlGaN/GaN heterostructure epitaxial wafer, as shown in Figure 2b. The 2DEG layer carrier concentration obtained by Hall measurement at room temperature was $9.7 \times 10^{12}$ cm$^{-2}$, and the electron mobility was 1541 cm$^2$ V$^{-1}$ s$^{-1}$.

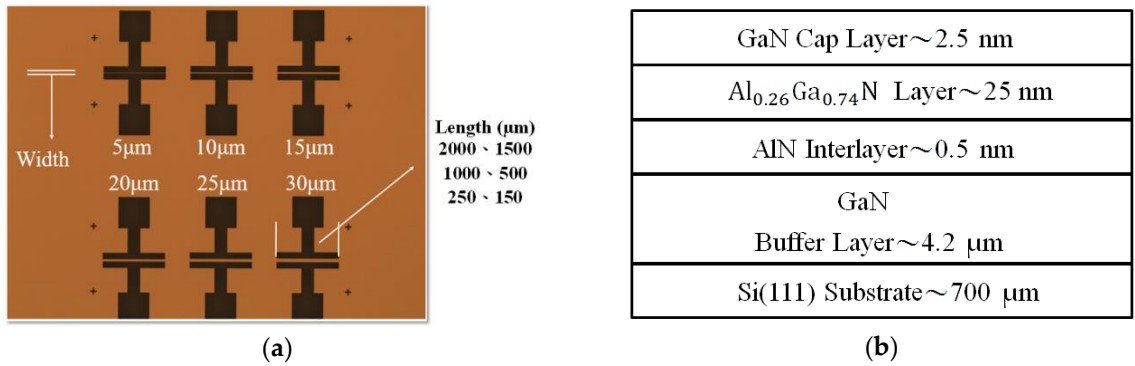

(**a**)  (**b**)

**Figure 2.** (**a**) Mask of the MSM varactors and (**b**) epitaxial structure of the AlGaN/GaN wafers.

Initially, all of the wafers were cut into a $1 \times 1$ cm$^2$ area by laser cutting. They were then subjected to a standard RCA cleaning procedure in an ultrasonic bath with acetone, isopropanol, and deionized water for three minutes each, sequentially. This was followed by spin-coating the S1813 photoresist on top of the wafers. An exposure machine was then used to complete the lithography process. Subsequently, an electron-beam evaporation machine was used to deposit the Ni/Au (20 nm/70 nm) Schottky junction metal layer. Finally, all the wafers were subjected to a lift-off procedure to obtain MSM varactors of different dimensions (Figure 3). In addition to the normal-sized varactors mentioned above, we reduced the sizes of all varactors to one-fifth by the exposure-scaling method. Next, to investigate the best component structure characteristics of the varactors, the capacitance–voltage (C–V), current–voltage (I–V), and breakdown characteristics of all these varactors were investigated.

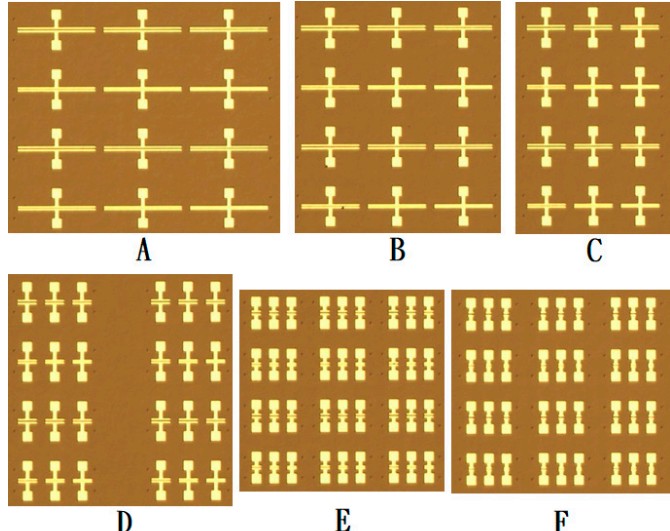

**Figure 3.** Diagrams of the final completed MSM varactors. From (**A**–**F**), the lengths are 2000, 1500, 1000, 500, 250, and 150 μm.

The measurements were made using an Agilent E4980A LCR-meter (Agilent Technologies, Santa Clara, CA, USA) and a Keithley 2410 SourceMeter (Keithley Instruments, Solon, OH, USA) to obtain the curves of electrical characteristics. The voltage measurement range of the C–V characteristic curve was from −15 to 15 V, measured once for every 0.05 V interval, and the frequency of the AC measurement signal was set to 2 MHz, 1 MHz, and 500, 200, 100, 50, and 10 kHz. The voltage measurement range of the I–V characteristic curve was done from −100 to 500 V, and the maximum current was limited to 21 mA because of equipment constraints.

## 2.2. Surge-Protection Circuit Design and Measurement

After completion of the experiment on the variabilities for the GaN-based MSM varactor and breakdown voltage, we used the design with an electrode length of 2000 μm and a gap width of 30 μm to complete the varactors. The wafer was then cut and bare-died by laser for use in the signal transmission and anti-surge protection circuit, as shown in Figure 4a. For the anti-surge module, we designed the 50 Ω microstrip line on the glass-reinforced epoxy laminate material (FR4) to be the signal transmission path. The GaN-based MSM varactor was connected in series between the microstrip line by flip-chip method with silver glue, and the low-parasitic-capacitance GDT was shunted in front of the varactor. The overall anti-surge module was completed as shown in Figure 4b. Surge current pulse injection measurements were performed, as shown in Figure 4c. Finally, the robustness of the anti-surge module was checked again using a network analyzer.

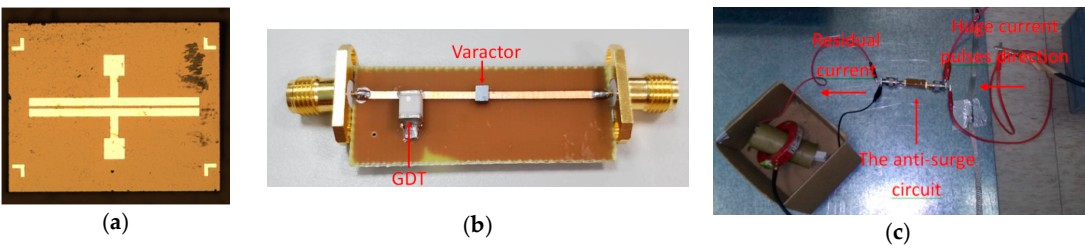

**Figure 4.** (**a**) The bare-die image of the GaN-based MSM varactor, (**b**) the overall anti-surge module with a GaN-based MSM varactor along with a gas discharge tube (GDT), and (**c**) the surge current pulse injection for residual current measurement (at 50 Ω load).

## 3. Results and Discussion

The three-dimensional structure of the MSM varactor is shown in Figure 5. The MSM varactor can be regarded as a double-gate (G) structure with two back-to-back Schottky junction diodes. When the bias voltage is zero or less than the threshold value, we can obtain a large capacitance value when measuring the C–V characteristic curve. However, when the bias voltage exceeds the threshold voltage, the capacitance of the varactor suddenly drops to an extremely small value. This is in accordance with the defining characteristic of varactors: their capacitance changes with the applied voltage [19].

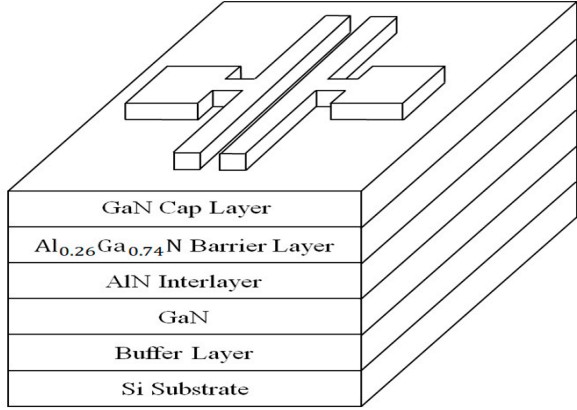

**Figure 5.** Three-dimensional structure of the MSM varactor.

### 3.1. Capacitor Characteristics and Breakdown Voltages of the Normal-Sized MSM Varactors

In this study, six electrode sizes for the MSM varactors were considered. The calculated areas, from large to small (Figure 3A–F), were 375,000, 325,000, 275,000, 225,000, 170,000, and 160,000 $\mu m^2$. The electrode area, low-voltage capacitance, high-voltage capacitance, and the capacitance conversion ratio are summarized in Table 2.

**Table 2.** Capacitance measurement values of MSM varactors with different electrode sizes.

| L (μm) | Area (μm²) | $C_{max}$ (pF) | $C_{min}$ (pF) | $C_{ratio}$ ($\frac{C_{max}}{C_{min}}$) |
|---|---|---|---|---|
| 2000 | 375,000 | 283 | 3.98 | 71 |
| 1500 | 325,000 | 252 | 3.59 | 70 |
| 1000 | 275,000 | 202 | 2.94 | 68 |
| 500 | 225,000 | 161 | 2.38 | 67 |
| 250 | 170,000 | 124 | 1.86 | 66 |
| 150 | 160,000 | 118 | 1.81 | 65 |

From the experimental results, it can be seen that the larger the electrode area was, the higher the capacitance was. In addition, the component length of 2000 μm gave the highest capacitance conversion ratio. We further measured the C–V and I–V characteristics of this design for different frequencies and gap widths. All of the results are given in Figures 6 and 7. The obtained maximum capacitance value of the varactor was 283 pF, and the maximum breakdown voltage was 390 V. In contrast to conventional metal–oxide–semiconductor field-effect transistor (MOSFET) devices, the gate has a breakdown voltage of only approximately 30 V. As a component, the breakdown voltage of varactors in this study was much higher.

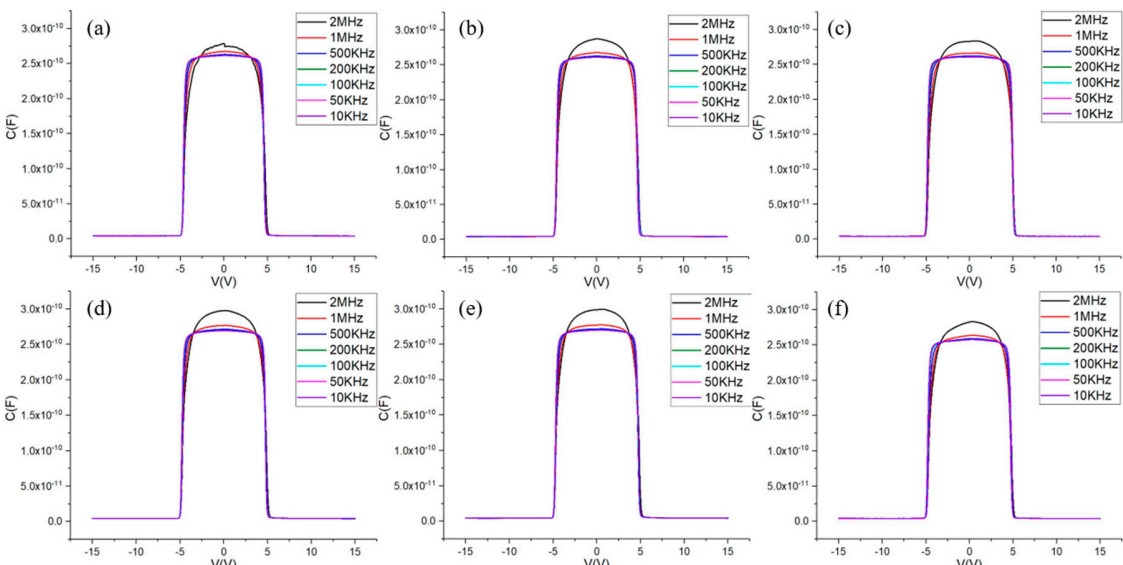

**Figure 6.** The C–V measurements of the MSM varactors with the same length (2000 μm) and different widths (from (**a**–**f**): 30, 25, 20, 15, 10, and 5 μm).

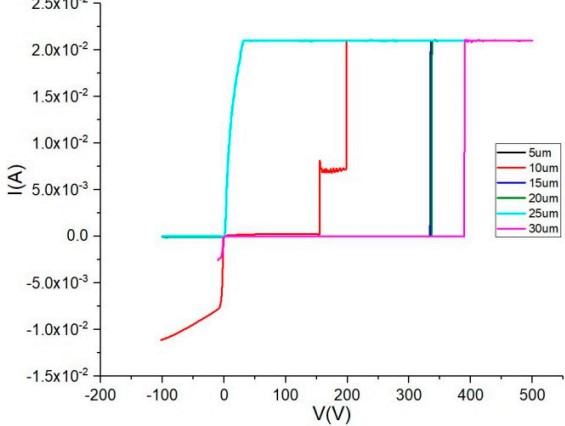

**Figure 7.** The I–V measurements of the MSM varactors with the same length (2000 μm) and different widths (the same as those in Figure 6).

### 3.2. Capacitor Characteristics and Breakdown Voltages of the Reduced-Size MSM Varactors

With the same manufacturing parameters, we further reduced the all dimensions of the normal-sized varactors by a factor of five. The dimensions of the reduced-size varactors are shown in Table 3. Since the varactor with a length of 2000 μm and a gap width of 30 μm can achieve the maximum capacitance and best capacitance conversion ratio, we chose the varactor that was shrunk from this design (reducing the length to 400 μm and the width to 6 μm) as a reduced-size sample for measuring the capacitance and breakdown voltage. The maximum capacitance of this varactor is 10.2 pF, the minimum is 0.589 pF, and the capacitance conversion ratio ($C_{max}/C_{min}$) is 17.3. The measurement data, given in Figure 8, show that the maximum capacitance of the reduced varactor decreases by a factor of 30 as compared with the normal size. This also explains why, after reducing the electrode to one-fifth of its size, the overall electrode area decreases from 375,000 to 15,000 μm². According to the capacitance formula, $C = \varepsilon A/d$, the values of $\varepsilon$ and $d$ are constant, but as the electrode area A becomes smaller, the corresponding overall capacitance value C decreases. In addition, compared with the normal-sized varactors, the capacitance of the reduced-size varactor becomes more stable as the frequency of the measuring signal changes (as shown in Figure 8). This phenomenon is worth further investigation and discussion.

**Table 3.** Dimensions of the reduced-size varactors.

| Normal Size | | Reduced Size to One-Fifth | |
|---|---|---|---|
| Length (μm) | Gap (μm) | Length (μm) | Gap (μm) |
| 2000 | 30 | 400 | 6 |
| 1500 | 25 | 300 | 5 |
| 1000 | 20 | 200 | 4 |
| 500 | 15 | 100 | 3 |
| 250 | 10 | 50 | 2 |
| 150 | 5 | 30 | 1 |

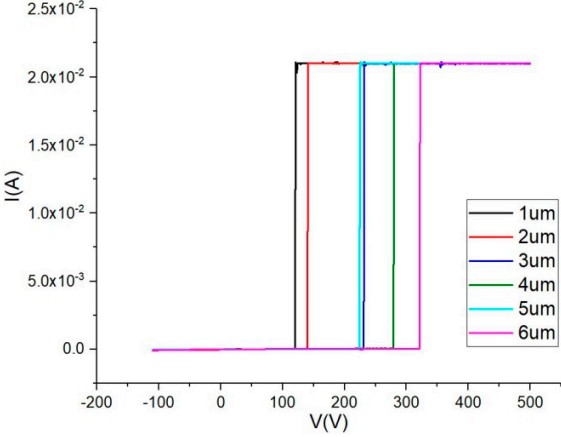

**Figure 8.** The C–V measurements of the MSM varactors with reduced size (length = 400 μm and gap width = 6 μm).

For the I–V characteristic curve measurement results, the breakdown voltages of the reduced-size varactors are shown in Figure 9. Among these, the varactor with a gap width of 6 μm achieves the highest breakdown voltage of 321 V.

**Figure 9.** I–V measurements of the MSM varactors with reduced size (length = 400 μm, different gap widths).

### 3.3. Breakdown Mechanism for MSM Varactors with Different Gap Widths

The measurement results obtained from Figures 7 and 9 show that the width of the electrode gap of the varactor greatly affects the breakdown voltage. Overall, the larger the gap width was, the higher the breakdown voltage was.

In order to further analyze the relationship between the electrode gap width and the breakdown voltage, we measured the I–V characteristic curve of the normal-sized varactor (length of 500 μm and

gap widths of 30, 25, 20, 15, 10, and 5 µm), as shown in Figure 10. Next, combining these measurement results with the reduced-size varactor (length of 400 µm and widths of 1, 2, 3, 4, 5, and 6 µm; shown in Figure 9), we can obtain the breakdown voltage relationship between the electrode gap width from 1 to 30 µm based on the similar length value, as shown in Figure 11.

We can see from this measurement result that, for an electrode gap width less than 10 µm, the slope of the breakdown voltage curve of the varactor is positive. This means that as the electrode gap width becomes larger, the breakdown voltage of the varactor increases. Under this small gap width condition, the bias voltage causes an extremely high electric field within the anode and cathode. This high electric field causes the surface material, the GaN cap layer, to break down and provide a path for the leakage current (as shown in Figure 11, path A). Conversely, because the electrode gap width is large (greater than 10 µm), the slope of the breakdown voltage curve of the varactor is nearly zero. This means that the electrode gap width will no longer be the main condition influencing the breakdown voltage. Although the electrode gap width becomes larger, the breakdown voltage remains at approximately 300 V. At this stage, the GaN cap layer on the surface can sustain a smaller electric field without breakdown, and so the breakdown path exists under the epitaxial wafer (as shown in Figure 11, path B). Consequently, the epitaxial thin-film quality of the GaN wafer becomes the most important condition for increasing the breakdown voltage.

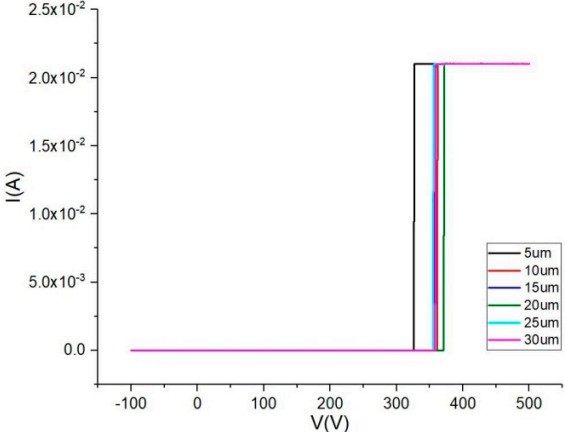

**Figure 10.** I–V measurements of the MSM varactors with the same length (500 µm) and different widths (the same as those in Figure 6).

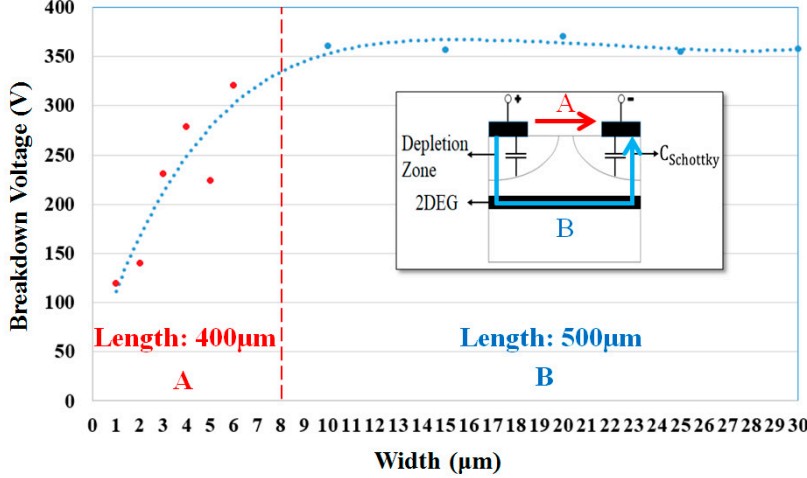

**Figure 11.** Breakdown mechanism of the MSM varactor with different gap widths (1 to 30 µm).

### 3.4. The Anti-Surge Module Application

Since the capacitance of MSM varactors changes with the applied bias voltage, when an MSM varactor is connected in series in the signal transmission path, the voltage of the signal transmission range is low under normal working conditions. Therefore, the high capacitance of the varactor offers low impedance to the signal. However, when facing strong high-voltage pulse injection, the capacitance of the MSM varactor rapidly decreases because of the high bias voltage. Therefore, the surge signal cannot couple through the transmission path, thereby protecting the back-end circuit module. This prevents ESD and MEMP surges from directly affecting the circuit and causing permanent damage. In this way, damage to the back-end components can be suppressed or blocked.

The C–V measurement results for a varactor with an electrode length of 2000 μm and a gap width of 30 μm are shown in the Figure 12a. The frequency of the measurement signal varies from 1 kHz to 2 MHz. The maximum capacitance ($C_{max}$) of the varactor is 455 pF, the minimum capacitance ($C_{min}$) is 5.08 pF, and the capacitance conversion ratio ($C_{ratio}$) can be up to 89.5. In addition, as described in the literature, it can be observed that the capacitance of the varactor is dependent on the measurement signal frequency. As the frequency decreases, the C–V curve assumes a "batman-like" curve. The results of network analyzer measurement of the overall anti-surge module are shown in Figure 12b. The insertion loss of the module is approximately −2 dB within the GPS devices' operating band (1.2 to 1.6 GHz). This proves that under normal operation, the capacitance is large enough, and therefore the signal transmission is still very efficient.

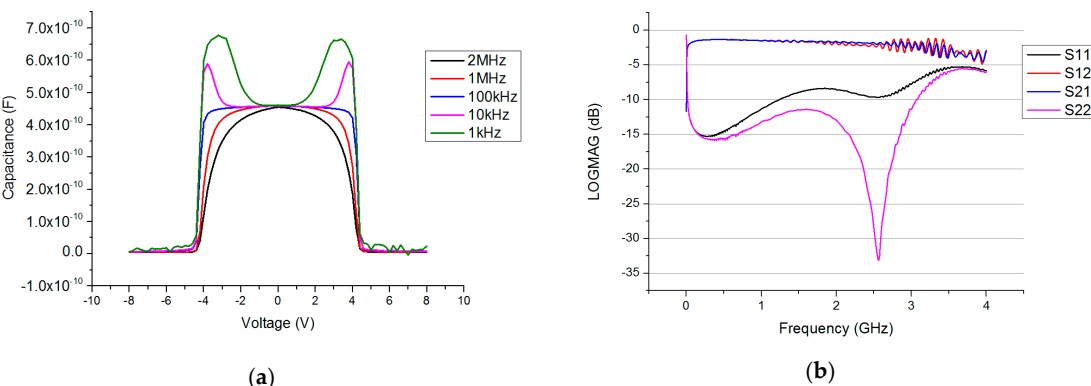

**Figure 12.** (**a**) C–V measurement results for a varactor with electrode length of 2000 μm and a gap width of 30 μm; (**b**) network analyzer measurement result for the overall anti-surge module.

The anti-surge module application was used next. The standard injection current pulses are shown in Figure 13a. According to the testing requirements, the current specifications can be performed from 600 A to 2.5 kA. As per the MIL-STD-188-125-2, with a 50 Ω dummy load resistor and a 600 A injected current pulse, all of the residual current value can be suppressed to less than 5 A. The best result is 3.84 A, as shown in Figure 13b. Furthermore, the network analyzer measurement is performed again to check whether or not the anti-surge module has been damaged. All of the S-parameters show that after 600 A current pulse injection, the module functionality was still significantly normal, as shown in Figure 14. In addition, all of the test results show that this anti-surge module can effectively suppress the surge energy as the injection current value keeps increasing up to 2.54 kA. The residual current values are not only less than 10 A, as shown in Figure 15a, but the S-parameters show no significant variation, as shown in Figure 15b. Hence, the GaN-based MSM varactor can be used in signal transmission paths; when the surge pulse attack occurs, the MSM varactor can block the surge energy in time because of its excellent capability to withstand voltage and because of its fast response abilities; hence, the GDT has sufficient time to start up and shunt, and the bulk of the surge energy can be discharged.

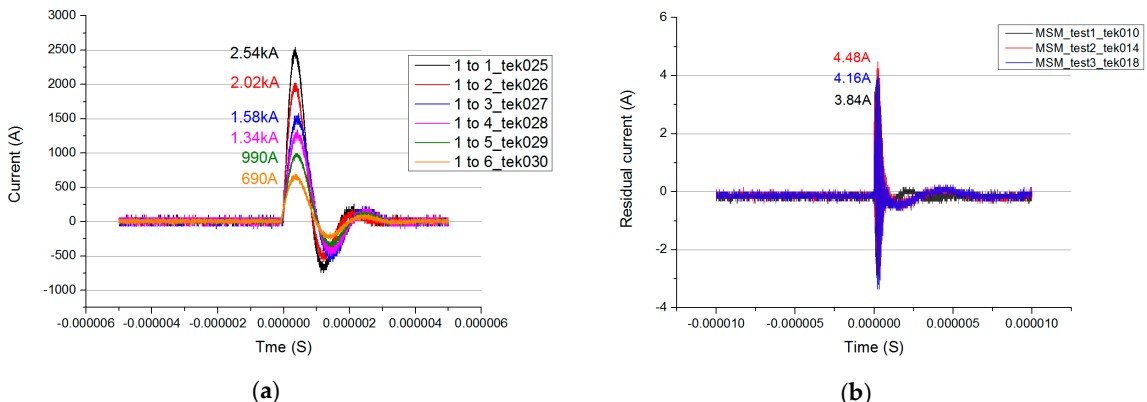

**Figure 13.** (**a**) Standard injection current pulses; (**b**) residual current values of the anti-surge module (with 50 Ω dummy load resistor and 600 A injected current pulse).

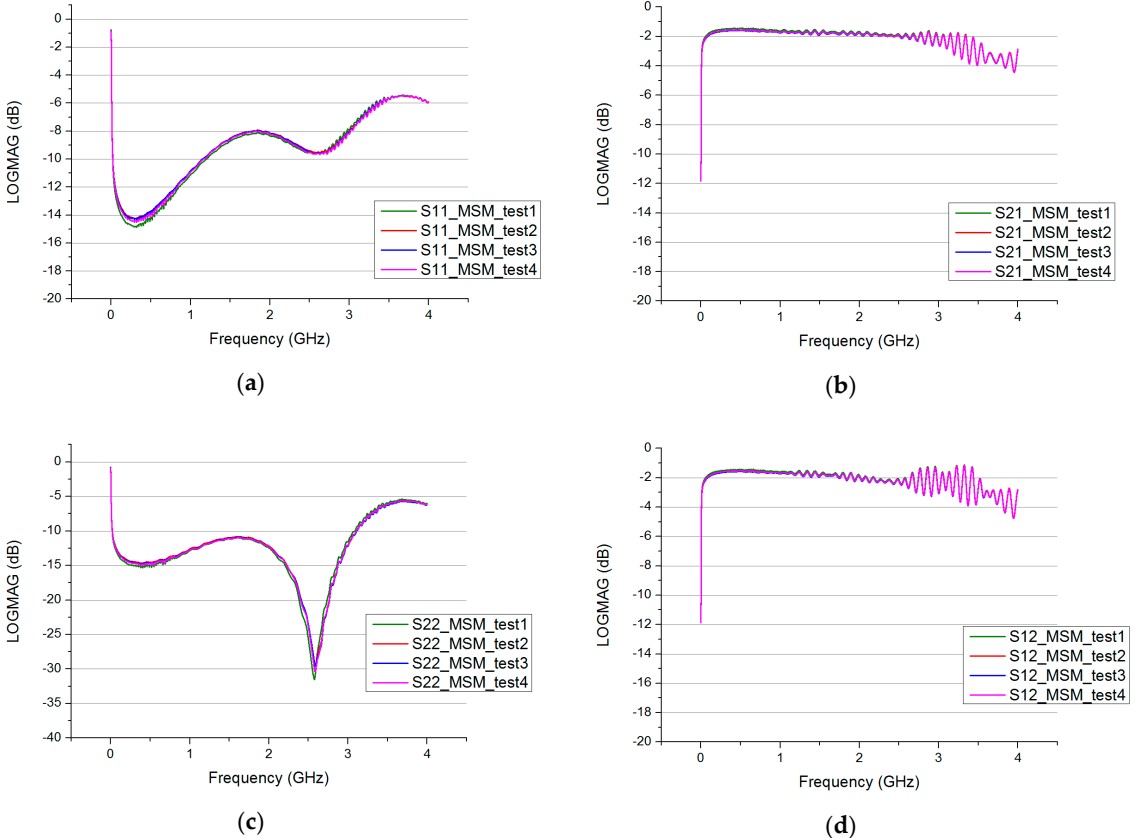

**Figure 14.** Network analyzer measurement results for the anti-surge module after injection of 600 A surge current for (**a**) input port voltage reflection coefficient (S11), (**b**) forward voltage gain (S21), (**c**) output port voltage reflection coefficient (S22), and (**d**) reverse voltage gain (S12).

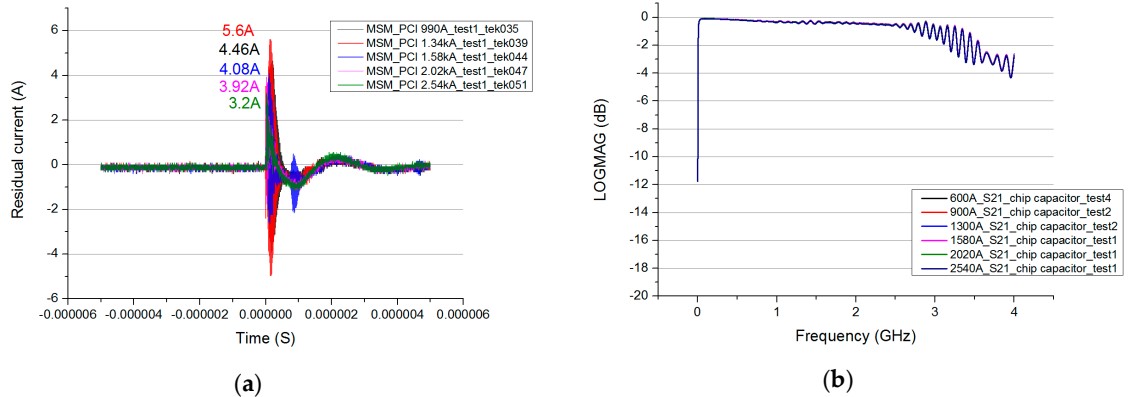

**Figure 15.** (**a**) Residual current of the anti-surge module (at 50 Ω and injection currents of 900 A to 2.54 kA); (**b**) S21 measurement results of the anti-surge module after injection of 900 A to 2.54 kA of the surge current.

## 4. Conclusions

We have fabricated GaN/AlGaN MSM varactors as surge protection components in this study for their suitable material properties. Accordingly, different electrode patterns of MSM varactors were designed, and the effects on their C–V and I–V characteristics were investigated.

Experimental results show that the maximum capacitance was 283 pF, the capacitance conversion ratio ($C_{ratio}$) was 71, and the breakdown voltage was as high as 390 V for an electrode length of 2000 μm and an electrode gap width of 30 μm. We found that when the electrode gap width was less than 10 μm, surface breakdown phenomena occurred or dominated and the correspondent breakdown voltage was proportional to the electrode gap width. Conversely, under path breakdown occurred as the electrode gap width exceeded 10 μm. At this condition, the epitaxial layers' quality and thickness of the GaN wafer became important conditions for increasing the breakdown voltage of varactors.

The anti-surge module application with the GaN-based MSM varactor placed in series with the signal transmission path (the arrangement exhibited a low insertion loss of −2 dB) suppressed a 2.54 kA injection surge current to a value less than 10 A. This meets the requirements of MIL-STD-188-125-2 and can effectively protect the back-end RF circuit from damage.

**Author Contributions:** Conceptualization, L.-B.C. and M.-J.J.; data curation, S.-A.H., Y.-L.H., and Z.-X.D.; funding acquisition, L.-B.C.; investigation, S.-A.H., Y.-L.H., C.-F.S., Z.-X.D., and L.-B.C.; methodology, L.-B.C., M.-J.J., and C.-F.S.; project administration, L.-B.C. and M.-J.J.; resources, L.-B.C. and M.-J.J.; supervision, L.-B.C.; writing of original draft, Y.-L.H., S.-A.H., C.-F.S., and L.-B.C.; writing of review and editing, C.-F.S., Y.-L.H., L.-B.C., and M.-J.J. All authors have read and agreed to the published version of the manuscript.

**Funding:** This research was funded by Chang Gung Memorial Hospital (CMRPD2I0062) and Ministry of Science and Technology, Republic of China (MOST108-2221-E-182-022).

**Acknowledgments:** The authors would like to thank Green Technology Research Center of Chang Gung University, Ministry of Science and Technology (R.O.C.), Chang Gung Memorial Hospital and Taiwan Semiconductor Research Institute for supporting this study.

**Conflicts of Interest:** The authors declare no conflict of interest.

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
