# Peer review of "Capacitance Characteristics and Breakdown Mechanism of AlGaN/GaN Metal–Semiconductor–Metal Varactors and their Anti-Surge Application"

_crystals, doi:10.3390/cryst10040292_

Round 1
Reviewer 1 Report
This work presents the design and fabrication of GaN/AlGaN MSM varactors for its use in surge protection circuits for RF applications. The effects of the device dimensions on its performance were accurately investigated and the signal transmission performance and anti-surge ability were presented. The results are clearly presented and adequately analyzed. Thus, I recommend its publication in the present form.
Reviewer 2 Report
Crystals: Manuscript Number: # crystals-744275v1, “Capacitor Characteristics and Breakdown Mechanism of 2 AlGaN/GaN Metal–Semiconductor–Metal Varactors and 3 their Anti-surge Application”, by Chien-Fu Shih, Yu-Li Hsieh, Liann-Be Chang, Ming-Jer Jeng, Zi-Xin Ding, Shao-An Huang, studied the Breakdown Mechanism of AlGaN/GaN Metal–Semiconductor–Metal Varactors and 3 their Anti-surge Applications. The manuscript has the worth of publication in Crystals and should be accepted after few clarifications or should be added in the revised manuscript as mentioned below:
- Authors should check the English syntax errors through the manuscript.
- Authors should improve the way of presentation specially the graphs.
Author Response
The manuscript has the worth of publication in Crystals and should be accepted after few clarifications or should be added in the revised manuscript as mentioned below:
- Authors should check the English syntax errors through the manuscript.
1 answer:Thank you for your comment. Our manuscript has been substantially revised by native English speakers.
2. Authors should improve the way of presentation specially the graphs.
2 answer:Thank you for your comment. In order to make all the graphics easier to read, we have modified the way to present the graphs.

Reviewer 3 Report
The idea and result are important to the readers.
The title can be changed: "Capacitance Characteristics and Breakdown Mechanism of" is not essential.
I think they can make their result and discussion part more compact.
1) In most of figures, the figure and font sizes are too small in printing.
2) Figure 3 is not essential.
3) In explaining Figure 5, "Compared with conventional HEMTs that have source (S), drain (D), and gate (G) terminals" is not necessary.
4) They explained the data long, yet, it is not easy to catch the importance of them.
For example, Table 2. can be simply shown in graphs.
Reviewer 4 Report
This manuscript describes an interesting approach for shielding a system against huge current pulses. It sounds very conclusive, although my expert knowledge about the topic is fairly limited. I do not understand why here a HEMT structure is necessary. Is it the excellent conductivity of the 2D channel?
Let me anyway ask some questions and give some comments:
1) Can the authors give some references from where the data in Tab. 1 are taken? I wonder particularly about the indicated electron and hole mobility of GaN.
2) Besides being unreadably small, it remains somewhat unclear what Fig. 2 presents or helps for understanding. What does it mean to have two parallel branches from the antenna to the system?
3) It is unclear whether figs. 4b and 4c are meaningful. Also here the question: What is the role of the GDT in these figures?
4) The capacitance conversion ratios in Tab. 2 last column are fairly the same. Does it make sense to discuss differences or is this just noise fluctuations?
5) Page 6 1st line: Statements like "the larger was the electrode area, the higher was the capacitance" are not really helpful. What else did you expect?
6) Similarly in the last line on page 6: Largest capacitance is a consequence of largest area (so why not choosing an even larger area?), whereas the conversion ratio is not really different to other sizes (see my point 4).
7) Why does the capacitance of some varactors depend on the frequency, why are small varactors more stable (page 7 line 180)? What is the reason for the batman-like shape in Fig. 12a?
8) Page 7 line 187: Again a quite trivial statement: The widest gap provides highest break-down voltage. What else is expected? See also page 8 line 194.
9) Fig. 12: What means "before flip-chip" and "before PCI"?
10) How can a well-defined current pulse be applied? Current is (for my understanding) a consequence of voltage.
11) In the conclusions, the authors mention that "the epitaxial thin-film quality and thickness of the GaN wafer became the most important conditions for increasing the breakdown voltage of varactors". I did not see respective investigations described in the main part.
12) Most figures and their labels etc. are too small, nearly unreadable.
